# Cuban Sugar Cane Wax Alcohol Exhibited Enhanced Antioxidant, Anti-Glycation and Anti-Inflammatory Activity in Reconstituted High-Density Lipoprotein (rHDL) with Improved Structural and Functional Correlations: Comparison of Various Policosanols

**DOI:** 10.3390/ijms24043186

**Published:** 2023-02-06

**Authors:** Kyung-Hyun Cho, Seung Hee Baek, Hyo-Seon Nam, Ji-Eun Kim, Dae-Jin Kang, Hyejee Na, Seonggeun Zee

**Affiliations:** 1Raydel Research Institute, Medical Innovation Complex, Daegu 41061, Republic of Korea; 2LipoLab, Yeungnam University, Gyeongsan 38541, Republic of Korea

**Keywords:** HDL, high-density lipoproteins, apoA-I, apolipoprotein A-I, policosanol, sugar cane wax alcohol, zebrafish, embryo

## Abstract

Policosanols from various sources, such as sugar cane, rice bran, and insects, have been marketed to prevent dyslipidemia, diabetes, and hypertension by increasing the blood high-density lipoproteins cholesterol (HDL-C) levels. On the other hand, there has been no study on how each policosanol influences the quality of HDL particles and their functionality. Reconstituted high-density lipoproteins (rHDLs) with apolipoprotein (apo) A-I and each policosanol were synthesized using the sodium cholate dialysis method to compare the policosanols in lipoprotein metabolism. Each rHDL was compared regarding the particle size and shape, antioxidant activity, and anti-inflammatory activity in vitro and in zebrafish embryos. This study compared four policosanols including one policosanol from Cuba (Raydel^®^ policosanol) and three policosanols from China (Xi’an Natural sugar cane, Xi’an Realin sugar cane, and Shaanxi rice bran). The synthesis of rHDLs with various policosanols (PCO) from Cuba or China using a molar ratio of 95:5:1:1 with palmitoyloleoyl phosphatidylcholine (POPC): free cholesterol (FC): apoA-I:PCO (wt:wt) showed that rHDL containing Cuban policosanol (rHDL-1) showed the largest particle size and the most distinct particle shape. The rHDL-1 showed a 23% larger particle diameter and increased apoA-I molecular weight with a 1.9 nm blue shift of the maximum wavelength fluorescence than rHDL alone (rHDL-0). Other rHDLs containing Chinese policosanols (rHDL-2, rHDL-3, and rHDL-4) showed similar particle sizes with an rHDL-0 and 1.1–1.3 nm blue shift of wavelength maximum fluorescence (WMF). Among all rHDLs, the rHDL-1 showed the strongest antioxidant ability to inhibit cupric ion-mediated LDL oxidation. The rHDL-1-treated LDL showed the most distinct band intensity and particle morphology compared with the other rHDLs. The rHDL-1 also exerted the highest anti-glycation activity to inhibit the fructose-mediated glycation of human HDL_2_ with the protection of apoA-I from proteolytic degradation. At the same time, other rHDLs showed a loss of anti-glycation activity with severe degradation. A microinjection of each rHDL alone showed that rHDL-1 had the highest survivability of approximately 85 ± 3%, with the fastest developmental speed and morphology. In contrast, rHDL-3 showed the lowest survivability, around 71 ± 5%, with the slowest developmental speed. A microinjection of carboxymethyllysine (CML), a pro-inflammatory advanced glycated end product, into zebrafish embryos resulted in severe embryo death of approximately 30 ± 3% and developmental defects with the slowest developmental speed. On the other hand, the phosphate buffered saline (PBS)-injected embryo showed 83 ± 3% survivability. A co-injection of CML and each rHDL into adult zebrafish showed that rHDL-1 (Cuban policosanol) induced the highest survivability, around 85 ± 3%, while rHDL-0 showed 67 ± 7% survivability. In addition, rHDL-2, rHDL-3, and rHDL-4 showed 67 ± 5%, 62 ± 37, and 71 ± 6% survivability, respectively, with a slower developmental speed and morphology. In conclusion, Cuban policosanol showed the strongest ability to form rHDLs with the most distinct morphology and the largest size. The rHDL-containing Cuban policosanol (rHDL-1) showed the strongest antioxidant ability against LDL oxidation, anti-glycation activity to protect apoA-I from degradation, and the highest anti-inflammatory activity to protect embryo death under the presence of CML.

## 1. Introduction

Policosanol is a mixture of aliphatic alcohols ranging from 24 to 34 carbon atoms [1,2], such as octacosanol, triacontanol, dotriacontanol, hexacosanol, and tetratriacontanol as the major components, which are purified from sugar cane (*Saccharum officinarum* L.) wax [1,2,3] or various plants, such as oats [4] and barley [5], insects [6,7], and bees wax [8]. Many policosanols have been purified from various plant sources, such as sugar cane, rice bran [9,10,11], wheat germ, and barley sprout. Despite this, no study has compared the policosanols among the many various sources and origins regarding the correlations between the chemical compositions and physiological effects, such as high-density lipoprotein (HDL)-binding ability and enhancement of HDL functionalities.

Many policosanols from different sources have been used to treat blood dyslipidemia, hypercholesterolemia, diabetes [11], hypertension [12,13], and dementia [7,14] by raising the HDL-C and lowering the LDL-C. However, except for Cuban policosanol [12,15,16], there is no sufficient information on policosanol about its physiological effects on lipoprotein metabolism, particularly in HDL functionality. In addition to the increase in HDL-C quantity, improvement of HDL quality and functionality should also be considered to maximize the efficacy of policosanol. The HDL quality in the blood and the antioxidant and anti-inflammatory properties may be improved by policosanol consumption [17,18] because dysfunctional HDL is more atherogenic and exacerbates the pro-inflammatory cascade [19].

This study compared the in vitro effects of many policosanols in terms of the HDL functionalities on the molecular level after synthesis with rHDL via the encapsulation of policosanol into rHDL particles, as reported previously [15,20]. A desirable policosanol should not interfere with normal HDL functionality, such as particle formation after uptake from the intestinal mucosal barrier via binding with apoA-I, antioxidant ability, and anti-inflammatory activity. The anti-glycation activity of HDL containing each policosanol (rHDL-PCO) has been evaluated by testing the fructose-mediated glycation with the rHDL, as reported previously [15,20]. Fructose, a ketohexose, induces glycation more rapidly than glucose [21] to produce advanced glycated end products (AGEs), which cause inflammation with neurotoxicity. Among the AGEs, an elevated serum *N*-ε-carboxymethyllysine (CML) level was also associated with the exacerbation of atherosclerosis via lipoprotein modifications and increased susceptibility of low-density lipoprotein (LDL) oxidation [22]. Higher serum levels of CML are associated with high-sensitivity C-reactive protein (CRP) via an increase in toll-like receptor 4 (TLR-4) expression in monocytes [23,24], suggesting that an elevated CML level is associated with a pro-inflammatory state [25].

The anti-inflammatory properties of various policosanols have been compared using zebrafish embryos by testing the developmental speed, swimming ability, and survivability after injection of the rHDL-PCO. The zebrafish (*Danio rerio*) is a widely used vertebrate model to test the putative anti-inflammatory effects of drug candidates because zebrafish have well-developed innate and acquired immune systems that are similar to the mammalian immune system [26]. An additional advantage of working with zebrafish embryos is that zebrafish embryos develop externally and are optically transparent during development. With these characteristics, zebrafish and their embryos are a useful and popular animal model for various studies, including inflammation [27].

This study compared physicochemical characterizations among four different policosanols after encapsulation of each policosanol, regarding particle size and shape, in the rHDL state. Each rHDL was evaluated in terms of structural and functional correlations, antioxidant, anti-glycation, and anti-inflammatory activity in vitro and in vivo using zebrafish embryos to provide information on the physiological potential of policosanol in HDL.

## 2. Results

### 2.1. Ingredients Composition Analysis

Four policosanols showed strikingly different total wax alcohol contents and compositions of eight long-chain aliphatic alcohols (C24–C34), as shown in Table 1. Cuban policosanol showed the largest total wax alcohol (mg/g) amount, more than 982 mg/g, while other policosanols showed 518–610 mg/g of total wax alcohol. Interestingly, Cuban sugar cane wax alcohol (policosanol 1) showed the optimal and desirable content of 1-octacosanol (mg/g), around 692 mg/g (~70%), in total wax alcohol. On the other hand, other policosanols showed various contents, 356 mg/g (~58%), 69 mg/g (~12%), and 492 mg (~95%), in total wax alcohol for policosanol 2, 3, and 4, respectively. These results suggest that the compositions of long-chain aliphatic alcohols may vary depending on policosanol products from various sources and countries of origin. Unexpectedly, the final total amount of ingredients in policosanol 3 and 4 were 15% and 47%, respectively, lower than that of the total amount on the label of each product (Table 1).

### 2.2. Synthesis of Reconstituted HDL with Policosanol

All policosanols showed sufficient binding ability with phospholipid and apoA-I to form rHDL (Table 2), exhibiting 4–5 nm blue-shifted wavelength maximum fluorescence (WMF) from 336.3 nm of lipid-free apoA-I, suggesting that intrinsic Trp 108 in apoA-I was moved to the nonpolar phase upon the binding of policosanol and apoA-I. An rHDL containing Cuban policosanol, rHDL-1, showed the largest particle size, around 75.1 nm in diameter, with 1.9 nm more blue-shifted WMF than rHDL-0. In contrast, other rHDLs showed a smaller diameter, approximately 56–63 nm, with a 1.1–1.3 nm blue-shifted WMF. After synthesis, each rHDL contained apoA-I (2 mg/mL) and designated policosanol (~25 μg/mL) after synthesis.

### 2.3. Electrophoretic Profiles of rHDL Containing Policosanol

Under a non-denatured state, native electrophoresis on agarose revealed that rHDL-1 showed the slowest electromobility among all rHDL-containing policosanols with distinct band intensity, as shown in Figure 1A. In contrast, rHDL-2, rHDL-3, and rHDL-4 showed similar electromobility with a smear band intensity. The differences may be due to different physicochemical properties of policosanol ingredients depending on product sources.

There was a difference in band mobility between rHDL-0 and rHDL-1; rHDL-0 showed a different single band intensity, while rHDL-1 showed split bands with a slightly smeared band intensity because of the presence of Cuban policosanol. Furthermore, the rHDL-1 showed slower electromobility with more distinct band intensity than rHDL-2, rHDL-3, and rHDL-4, suggesting a larger particle size and less oxidized extent in rHDL1.

Under the denatured state, SDS-PAGE revealed that apoA-I in rHDL-0 (lane 1) showed a similar band position and molecular weight with lipid-free apoA-I (28 kDa, lane 6), as shown in Figure 1B. On the other hand, apoA-I in rHDL-1 (lane 2) showed a higher band position and a higher molecular weight than apoA-I in rHDL-0, suggesting that the MW of apoA-I was increased slightly via putative binding with policosanol.

Transmission electron microscopy (TEM) showed that rHDL-1 showed the most distinct disc particle shape with rouleaux morphology and the highest particle number. In contrast, rHDL-2 showed the smallest particle number and severely unclear and ambiguous morphology, as shown in Figure 2. The rHDL-3 and rHDL-4 showed a smaller particle number than rHDL-0, even though rHDL-4 showed an unclear morphology with a more likely multilamellar vesicle shape. All rHDLs showed similar particle sizes of 56–64 nm, except rHDL-1, which was around 75 nm in diameter.

### 2.4. Anti-Glycation Activity

As shown in Figure 3A, a fructose treatment on human HDL_2_ caused severe glycation, with a six-fold higher yellowish fluorescence intensity (FI) than HDL_2_ alone during a 48 h incubation under 5% CO_2_. However, treatment of rHDL-1 caused remarkable inhibition of the HDL_2_ glycation, up to 25% lower glycation extent than HDL_2_ alone, at 48 h incubation, while other rHDLs showed no significant inhibition, approximately 4–15% inhibition.

As shown in Figure 3B, electrophoretic analysis with each HDL_2_ sample showed that HDL_2_ alone had a distinct apoA-I band (28 kDa). In contrast, glycated HDL_2_ (lane 2) showed a remarkably diminished apoA-I band with the protein aggregation in the start position of the running gel, as indicated by the red arrowhead. On the other hand, rHDL-1-treated HDL_2_ showed the strongest apoA-I band (28 kDa) without protein aggregation at the loading position. The apoA-I band of the other rHDLs virtually disappeared with severe protein aggregation. These results suggest that Cuban policosanol in rHDL-1 could inhibit the glycation of HDL_2_ and protect apoA-I from degradation in the presence of high fructose concentration (final 250 mM), whereas other policosanols did not show anti-glycation activity.

### 2.5. Inhibition of LDL Oxidation

Native LDL showed the strongest band intensity with the slowest electromobility (lane 1), as indicated by the dotted red line in Figure 4A, whereas oxidized LDL (Cu^2+^ treated) showed almost no band intensity with the fastest electromobility (lane 2). On the other hand, the co-treatment of rHDL-1 (final 200 μg of apoA-I and 3 μg of policosanol) resulted in a stronger band intensity and slower electromobility than those of rHDL-0, indicating that encapsulated Cuban policosanol exerted antioxidant activity to inhibit Cu^2+^-mediated LDL oxidation. The more oxidized LDL moved faster to the bottom of the gel, with more smear and a weaker band intensity due to the fragmentation of apo-B by oxidation. The other rHDLs did not exhibit potent inhibition activity; rHDL-2 showed adequate inhibitory activity to show a less smeared LDL band intensity, but an aggregated band appeared on the loading position, as indicated by the red arrowhead. In addition, more oxidized LDL resulted in protein aggregation in the loading position.

A determination of the oxidation extent by a TBARS assay showed that the oxidized LDL by the cupric ion treatment showed a 10-fold increase in malondialdehyde (MDA) content, as shown in Figure 4B. On the other hand, co-treatment of rHDL-1 resulted in a 35% decrease in MDA in LDL, whereas a co-treatment of rHDL-0 resulted in a 5% decrease in MDA. Interestingly, rHDL-2 and rHDL-4 exhibited adequate antioxidant activity, 15% and 18% lower MDA than oxLDL alone, while rHDL-3 had no effect on inhibiting LDL oxidation.

### 2.6. Embryo Survivability after Injection of Each rHDL

In order to compare the embryotoxicity of each policosanol, 20 nL of each rHDL (16 ng of apoA-I/250 pg of each policosanol) was microinjected into zebrafish embryos within four-hour post-fertilization (hpf). As shown in Figure 5A, during 24 h post-injection, the rHDL-0-injected embryo showed 73 ± 4% survivability, while the PBS-injected embryo also showed 82 ± 3% survivability with similar developmental morphology and speed. This result suggests no notable impairment to attenuate embryo development by a microinjection of rHDL, even though the survivability was lower than the PBS-injected embryo without significance. On the other hand, the rHDL-1-injected embryo showed higher survivability (~85 ± 3%) than that of rHDL-0, while each embryo injected with rHDL, rHDL-2, rHDL-3, and rHDL-4 showed lower survivability (62–79%). These results suggest that different policosanols in rHDL could influence different embryo survivability; rHDL-3, containing policosanol-3 from Chinese sugar cane, showed the lowest survivability (62 ± 6%).

Observation of the embryo morphology with a stereoimage showed that PBS-injected and rHDL-0-injected embryos showed a similar developmental speed and morphology. The rHDL-1-injected embryo showed the fastest developmental speed with distinct eye pigmentation and tail elongation at 24 h post-injection. At 48 h, all zebrafish hatched and swam except for the rHDL-2- and rHDL-3-injected embryos. In particular, the rHDL-3-injected embryos showed more deformity of the larvae and the highest number of unhatched embryos, as shown in Figure 5B. At 72 h post-injection, the rHDL-3-injected embryo showed the slowest developmental speed with the lowest survivability (photo e), whereas the rHDL-4-injected embryo showed a normal developmental speed and morphology. During 72 h, the rHDL-1-injected embryo showed the highest survivability and the fastest developmental speed, suggesting that Cuban policosanol exerted the highest protection ability of the embryo via stabilization of apoA-I.

### 2.7. Embryo Survivability after Co-Injection of Each rHDL and CML

As shown in Figure 6A, a microinjection of CML (500 ng) into zebrafish embryos resulted in the most severe death (30 ± 3% survivability) during 24 h, while PBS alone showed 83 ± 3% survivability. In the presence of CML, a co-injection of rHDL-1 resulted in the highest embryo survivability of approximately 86 ± 3%, whereas the co-injection of rHDL-0 resulted in a lower survivability of approximately 67 ± 7% (*p* = 0.002). The rHDL-2- or rHDL-3-injected embryos showed similar survivability (67 ± 5% and 63 ± 7%, respectively), while the rHDL-4-injected embryo showed the second highest survivability, approximately 71 ± 6% in the presence of CML. Although all rHDLs, with or without policosanol, showed potent anti-inflammatory activity against CML toxicity, rHDL-1 exerted the strongest anti-inflammatory activity to recover the highest survivability and fastest development. This result suggests that the anti-inflammatory activity of rHDL alone could be enhanced by incorporating Cuban policosanol. Interestingly, rHDL-3 showed the lowest survivability in the absence or presence of CML (Figure 5 and Figure 6), indicating that the quality of rHDL was affected by the type of encapsulated policosanol.

Observation of the embryo with a stereo image showed that the CML-alone-injected embryo exhibited the most severe embryonic defects, with retardation of developmental speed in eye pigmentation and tail elongation at the 21-somite stage, as shown in Figure 6B (photo b). The co-injection of rHDL-0 improved the normal developmental speed and morphology, but there were still unhatched embryos (~47%) at 48 h. On the other hand, the co-injection of rHDL-1 resulted in the most improved developmental speed and morphology; all embryos showed primordium-6 stage with the darkest eye pigmentation and tail elongation with more than 32 somites. At 48 h post-injection in the presence of CML, 67% of embryos in the rHDL-1 group hatched and showed normal swimming ability, whereas the PBS group showed unhatched embryos (~10%). Interestingly, the rHDL-2, rHDL-3, and rHDL-4 groups showed a much slower developmental speed than the rHDL-0 group, with the developmental morphological defects as indicated by the red arrowhead at 24 h (Figure 6B); rHDL-3 exhibited the slowest eye pigmentation speed and tail elongation, as indicated by the blue arrowhead in photo f in Figure 6B. At 48 h, almost all embryos in the rHDL-2, rHDL-3, and rHDL-4 groups were unhatched with no swimming ability. Furthermore, malformation of larvae appeared in the rHDL-2, rHDL-3, and rHDL-4 groups at 72 and 96 h. In particular, rHDL-3 showed the highest number of malformations with the lowest hatched ratio of approximately 20%. These results suggest that Cuban policosanols in rHDL contributed to the protection of embryos against CML-mediated embryotoxicity, while Chinese policosanol did not.

## 3. Discussion

Since the first report on policosanol in Cuba was published in 1993 [1], many policosanol products have been developed for global marketing, mainly claiming to treat dyslipidemia by lowering LDL-C and raising HDL-C, even though there are conflicting data depending on many sources, countries of origin, and brands of policosanol [29,30]. On the other hand, there has been a notable difference in the compositions of many policosanols depending on vegetable sources [31]. Interestingly, in 2002, Berthold’s group showed that policosanol is a promising phytochemical alternative for lipid reduction [32]. On the other hand, the same group reported that policosanol consumption had no lipid-lowering effects during a 12-week study in 143 hyperlipidemic patients [33]. Moreover, the publications have not defined the specific content ratios of the other policosanol products from outside of Cuba. Cuban policosanol was defined as genuine policosanol with a specific ratio of each ingredient [28]: 1-tetracosanol (C_24_H_49_OH, 0.1–20 mg/g); 1-hexacosanol (C_26_H_53_OH, 30.0–100.0 mg/g); 1-heptacosanol (C_27_H_55_OH, 1.0–30.0 mg/g); 1-octacosanol (C_28_H_57_OH, 600.0–700.0 mg/g); 1-nonacosanol (C_29_H_59_OH, 1.0–20.0 mg/g); 1-triacontanol (C_30_H_61_OH, 100.0–150.0); 1-dotriacontanol (C_32_H_65_OH, 50.0–100.0 mg/g); 1-tetratriacontanol (C_34_H_69_OH, 1.0–50.0 mg/g).

Although it is not easy to find what component is responsible for the best effect of Cuban policosanol, it has been suggested that the specific ratio of alcohol ingredients are more important to exert the activity [28]. From the determination of gas chromatography (Table 1), Cuban policosanol showed the highest total amount of aliphatic alcohols and 1-octacosanol content (692 mg/g, around 70% in total amount). However, future research is necessary to find what component and ratio are the best optimum to exert the beneficial activity.

This study compared the four policosanols regarding ingredient compositions of wax alcohols after gas chromatography analysis. As shown in Table 1, the four policosanols showed strikingly different distributions of long-chain alcohol distributions, indicating that they had different physicochemical properties in vitro in the rHDL state and physiological functions in vivo. Because policosanol is extremely insoluble in aqueous buffer, it has been difficult to compare its physiological activity in vitro with the quality of different policosanols from various sources. To overcome the hurdle, the policosanol was incorporated into rHDL with apoA-I to evaluate the physiological functions in lipoprotein metabolism, as in a previous report [15,20]. The four policosanols in the rHDL were tested after incorporating each policosanol with apoA-I because the apoA-I binding ability of policosanol is critical for rHDL formation and its structural and functional correlations [15,20]. In particular, the current results showed that each policosanol was easily incorporated into rHDL with different extents of blue-shifted WMF, particle size (Table 1), and electromobility (Figure 1). After synthesis of the rHDL, the molecular weight of apoA-I (28 kDa, lane 6) was increased slightly, around 30 kDa in rHDL-1, rHDL-2, rHDL-3, and rHDL-4 (lanes 2–5), while rHDL-0 showed 28 kDa (lane 1). Although it was very hard to detect apoA-I by Western blot due to a very low blotting efficiency, it was reasonable to visualize Coomassie blue staining because apoA-I was only one protein component in the rHDL.

These results indicate that each policosanol could bind with apoA-I to induce slower electromobility of apoA-I in the rHDLs, as the debris of the phospholipid mixture appeared in the bottom of the gel in Figure 1B, as indicated by the red arrowhead. The rHDL-1 showed slower electromobility (Figure 1A), a larger blue-shift of WMF with a larger particle size (Table 1), and a more distinct particle shape with a rouleaux morphology (Figure 2) than the other rHDLs, indicating that Cuban policosanol showed the highest putative hydrophobic interaction between the amphipathic helix domain of apoA-I. The highest interaction of policosanol and apoA-I was associated with the protection of the apoA-I band from proteolytic degradation of apoA-I via glycation stress (Figure 3) and prevention of LDL from the degradation of apo-B via oxidative stress (Figure 4). The enhanced stability of rHDL-1 also exhibited the highest survivability, with normal development speed and morphology of zebrafish embryos (Figure 5). In the presence of CML, co-injection of rHDL-1 showed the highest protective ability with the highest survivability and the fastest developmental speed (Figure 6).

The highest antioxidant ability and anti-glycation activity of rHDL-1 make a good agreement with a previous paper [15,17,18,20] and other reports [34]; rHDL-containing policosanol (final 10 μM) inhibited Cu^2+^-mediated LDL oxidation [15] and the susceptibility to LDL oxidation in vitro was reduced by 5–10 mg/day of Cuban policosanol for eight weeks [34]. Although Cuban policosanol (final 9.3 μM) alone exhibited adequate anti-glycation activity, the same concentration of policosanol in rHDL had a more potent anti-glycation activity [15]. In the same context, eight weeks of Cuban policosanol consumption (10 mg/day) resulted a remarkable decrease in the glycation extent of HDL_2_ by up to 22% in middle-aged male subjects [16] and women participants [17]. Furthermore, 24 weeks of Cuban policosanol consumption (10–20 mg/day) resulted in a remarkable decrease in LDL oxidation and HDL glycation in healthy subjects with prehypertension [18]. These results suggest that the in vitro potential of policosanol could be enhanced by incorporation into rHDL. The in vitro potentials are linked with the in vivo efficacy in a human clinical study.

Although rHDL-1 showed the best quality to exert the strongest antioxidant and anti-glycation activity with the highest embryo survivability, rHDL-3 showed the worst quality and the lowest embryo survivability in the absence or presence of CML (Figure 5 and Figure 6). In the same context, rHDL-3 did not inhibit the glycation of HDL_2_ (Figure 3) and oxidation of LDL (Figure 4) with the highest MDA level. Interestingly, rHDL-3 contained Chinese sugar cane policosanol (Xi’an Realin), which had the lowest 1-octacosanol (C28) content (~69 mg/g) and the highest 1-tetracosanol (C24) and 1-triacontanol (C33) contents (~56 mg/g and 236 mg/g, respectively). On the other hand, Cuban policosanol in rHDL-1 showed the highest octacosanol content (~692 mg/g) and the lowest tetracosanol content (~0.3 mg/g). These striking differences in wax alcohol compositions between rHDL-1 and rHDL-3 may affect the remarkable differences in the in vitro functionality and in vivo efficacy to exert anti-glycation (Figure 3), antioxidant (Figure 4), and anti-inflammatory activity (Figure 5 and Figure 6), as well as electromobility of the particle (Figure 1) and structure (Figure 2).

Because policosanol consists of long-chain aliphatic alcohols, which are extremely hydrophobic, each chain of the long-chain alcohols should be incorporated with a vesicle such as lipoprotein after intake. ApoA-I, a major protein of HDL, and is not only expressed in liver (hepatocytes) but also expressed in the intestine (enterocytes) [35], as a part of HDL or very-low-density lipoproteins (VLDL) in liver, and chylomicrons in the intestine. Although the precise mechanism is still unclear, it is reasonable to postulate that the ingredients of policosanol can be absorbed via binding with lipoprotein-like vesicles from the intestine.

The binding ability of policosanol with apoA-I for discoidal rHDL formation is very important for exerting physiological activities by maximizing the pluripotent functionality of HDL to prevent atherosclerosis, dyslipidemia, hypertension, and dementia. This is because lipid-free apoA-I and apoA-I-rHDL with a disc shape can cross the blood–brain barrier to bind with β-amyloid (Aβ) and inhibit the aggregation of amyloid plaques in the brain side [36]. On the other hand, healthy HDL has anti-infection activities to kill SARS-CoV-2 with cytoprotective activity, while glycated HDL loses the antiviral activity and is more cytotoxic to host cells [37]. Glycated apoA-I and reconstituted HDL has shown severe structural and functional modifications to accelerate atherosclerosis and senescence [21,38]. Many beneficial effects of HDL could be impaired by undesirable modifications, such as oxidation and glycation, to produce dysfunctional HDL, which has more pro-atherogenic and pro-inflammatory properties. Patients with diabetes mellitus or hypertension are more sensitive to COVID-19 infection and have a higher risk of mortality [39] because their HDL-C levels are remarkably decreased [40].

As summarized in Figure 7, rHDL containing policosanol (Raydel^®^), rHDL-1, showed the bigger particle size and more particle numbers than other rHDLs (Figure 1 and Figure 2) and displayed anti-glycation activity to protect apoA-I (Figure 3), and antioxidant activity to protect LDL (Figure 4). These activities of the rHDL-1 were linked with protectional activity of zebrafish embryos via anti-inflammatory activity against CML toxicity and inhibition of toll-like receptor (TLR)-2/TLR-4 signaling. It has been proposed that the inhibiting of TLR signaling pathways is an effective therapeutic strategy for suppressing unwanted inflammatory cascades [41], especially AGE and CML. HDL-like nanoparticles could act as TLR-4 antagonists by sequestering lipopolysaccharide (LPS), indicating that HDL inhibits TLR-4 signaling [42]. Reconstituted HDL also exhibited an anti-inflammatory effect by inhibiting TLR-4 signaling and reducing TLR-4 expression [43]. Recently, a microinjection of CML caused acute embryo death with severe developmental defects and retardation [44], suggesting that the AGE, such as CML, could induce severe embryo death with inflammation. The current results showed that the toxicity of CML could be neutralized by co-injection of rHDL containing policosanol; especially, rHDL-1 exhibited the strongest inhibition ability, while other rHDLs did not. These results suggest that supplementation of Cuban policosanol may protect HDL functionality and maximize its antioxidant and anti-inflammatory activity because the glycation of high-density lipoproteins (HDL) is associated with the production of dysfunctional HDL [45]. These results may explain why Cuban policosanol showed potent efficacy to treat metabolic diseases, such as dyslipidemia, hypertension [12,17,18], and gastric cancer [46].

In conclusion, rHDL containing Cuban policosanol showed more improved structural and functional correlations than rHDL alone or containing other policosanols. The enhanced quality and functionality of the rHDL containing Cuban policosanol helped inhibit the oxidation of LDL, glycation of HDL, and inflammatory death of embryos.

## 4. Materials and Methods

### 4.1. Materials

Palmitoyloleoyl phosphatidylcholine (POPC, #850457) was supplied by Avanti Polar Lipids (Alabaster, AL, USA). Sodium cholate (#C1254) was procured from Sigma (St Louis, MO, USA). *N*-ε-carboxymethyllysine (CAS-No 941689-36-7, Cat#14580-5g) and fructose (CAS-No 57-48-7, Cat #F0127) were purchased from Sigma–Aldrich (St. Louis, MO, USA). Policosanol 1, sugar cane wax alcohol, was obtained from National Center for Scientific Research (CNIC), Habana, Cuba. Policosanols 2 and 3 were from sugar cane and supplied by Xi’an Natural Field Biotechnique Co., Ltd. (Xi’an, China) and Xi’an Realin Biotechnology Co., Ltd. (Xi’an, China), respectively. Policosanol 4, from rice bran, was purchased from Shaanxi Pioneer Biotech (Xi’an, China). All raw materials of each policosanol were analyzed using the same procedure by gas chromatography (HP-5890A GC, Agilent, Palo Alto, CA, USA) with a GC-flame ionization detector and a Zebron ZB-5 column (30 m × 0.53 mm × 1.50 μm) from Phenomenex (Torrance, CA, USA) at the Korea Advanced Food Research Institute (Uiwang-si, Republic of Korea). Certificates of analysis are available in the Appendix A.

### 4.2. Purification of Lipoproteins

LDL (1.019 < d < 1.063), HDL_2_ (1.063 < d < 1.125), and HDL_3_ (1.125 < d < 1.225) were isolated from the sera of young and healthy human males (mean age, 23 ± 2 years), who donated blood voluntarily after fasting overnight, by sequential ultracentrifugation. The protocol of human blood donation was conducted according to the guidelines of the Declaration of Helsinki and approved by the Institutional Review Board of Yeungnam University (approval code 7002016-A-2016-021, approval date 4 July 2016). The density was adjusted by adding NaCl and NaBr, as detailed elsewhere [47], and the procedures were carried out in accordance with the standard protocols [48]. The samples were centrifuged at 100,000× *g* for 24 h at 10 °C using a Himac CP100-NX with a fixed angle rotor P50AT4 (Hitachi, Tokyo, Japan) at the Raydel Research Institute (Daegu, Republic of Korea). After centrifugation, each lipoprotein sample was dialyzed extensively against Tris-buffered saline (TBS; 10 mM Tris-HCl, 140 mM NaCl, and 5 mM ethylene-diamine-tetraacetic acid (EDTA) [pH 8.0]) for 24 h to remove the NaBr.

### 4.3. Purification of Human apoA-I

ApoA-I was purified from HDL by ultracentrifugation, column chromatography, and organic solvent extraction using the method described by Brewer et al. [49]. At least 95% protein purity was achieved, as confirmed by SDS-PAGE.

### 4.4. Synthesis of Reconstituted HDL

Reconstituted HDL (rHDL) was prepared using the sodium cholate dialysis method [50] at an initial molar ratio of 95:5:1:0 and 95:5:1:1 for POPC:cholesterol:apoA-I:policosanol, respectively. After dialysis, all rHDLs showed a similar range of residual endotoxin levels, 3.1–3.3 EU/mL, based on endotoxin quantification using a commercially available test kit (BioWhittaker, Walkersville, MD, USA).

### 4.5. Protein Determination

Protein content of purified HDL and LDL from ultracentrifugation and reconstituted HDL were analyzed by Lowry assay, which was modified by Markwell et al. [51], using bovine serum albumin (BSA) as a standard. Lipid-free apoA-I was determined by Bradford protein assay kit (Quick Start ™ Bradford Protein Assay Kit, Bio-Rad #5000201) using BSA as a standard.

### 4.6. Comparison of Electromobility

Each rHDL was subjected to agarose electrophoresis to compare electromobility in the native state. Native electrophoresis was carried out with a non-denatured rHDL sample to compare electromobility on 0.6% agarose gel depends on the three dimensional structure of apoA-I/HDL and oxidation extent. In order to keep the native state, the agarose gel had no SDS and electrophoresis was carried without SDS treatment and boiling of the sample. After running, the gel was dried under vacuum at 37 °C. After, apoA-I bands were visualized by Coomassie brilliant blue staining (final 1.25%).

### 4.7. Characterization of Trp Fluorescence in the rHDL

The wavelengths of maximum fluorescence (WMF) of the tryptophan (Trp) residues in apoA-I, in the lipid-free and lipid-bound states, were determined from the uncorrected spectra using an FL6500 spectrofluorometer (Perkin-Elmer, Norwalk, CT, USA) with Spectrum FL software version 1.2.0.583 (Perkin-Elmer), as described elsewhere [52], using a 1-cm path-length Suprasil quartz cuvette (Fisher Scientific, Pittsburgh, PA, USA). The samples were excited at 295 nm to avoid tyrosine fluorescence, and the emission spectra were scanned from 305 to 400 nm at room temperature.

### 4.8. Oxidation of LDL

Oxidized LDL (oxLDL) was produced by incubating the LDL fraction with CuSO_4_ (Sigma # 451657) at a final concentration of 10 μM for 4 h at 37 °C. The OxLDL was then filtered (0.22-μm filter) and analyzed using a thiobarbituric acid reactive substances (TBARS) assay to determine the extent of oxidation with a malondialdehyde (MDA, Sigma # 63287) standard, as described previously [53].

Under presence of Cu^2+^ in LDL, the antioxidant ability of each rHDL was tested by comparison of electromobility using 0.5% agarose gel, as described previously [54]. Comparison of relative electromobility of a mixture of LDL (8 μg of protein) and each rHDL (0.5 μg of protein) was carried out under a non-natured state on 0.5% agarose gel (120 mm length × 60 mm width × 5 mm thickness). The electrophoresis was carried out with 50 V for 1 h in Tris-acetate-EDTA buffer (pH 8.0). The apo-B in LDL was visualized by Coomassie brilliant blue staining (final 1.25%). More oxidized LDL was moved faster to the bottom of the gel due to apo-B fragmentation and increase of negative charge.

### 4.9. Electron Microscopy

Transmission electron microscopy (TEM, Hitachi, model HT-7800; Ibaraki, Japan) was performed at 80 kV at the Raydel Research Institute (Daegu, Republic of Korea). HDL3 was stained negatively with 1% sodium phosphotungstate (PTA; pH 7.4) with a final protein concentration of 0.3 mg/mL in TBS. A 5 μL aliquot of the HDL suspension was blotted with filter paper and replaced immediately with a 5 μL droplet of 1% PTA. After a few seconds, the stained HDL fraction was blotted onto a Formvar carbon-coated 300 mesh copper grid and air-dried. The shape and size of the HDL were determined by TEM at a magnification of 40,000×, according to previous reports [55].

### 4.10. Glycation of HDL_2_ under the Presence of rHDL 

The glycation sensitivity was compared by incubating the purified lipid-free apoA-I (final 1 mg/mL) with 250 mM D-fructose (Sigma # F2793) in 200 mM potassium phosphate/0.02% sodium azide buffer (pH 7.4), as reported elsewhere [21,37]. ApoA-I was incubated for up to 48 h in an atmosphere containing 5% CO_2_ at 37 °C. The extent of the advanced glycation reactions was determined by reading the fluorescence intensities at 370 nm (excitation) and 440 nm (emission), as described previously [56], using an FL6500 spectrofluorometer (Perkin-Elmer, Norwalk, CT, USA) with Spectrum FL software version 1.2.0.583 (Perkin–Elmer) and a 1 cm path-length Suprasil quartz cuvette (Fisher Scientific, Pittsburgh, PA, USA).

### 4.11. Zebrafish Maintenance

Zebrafish and embryos were maintained using the standard protocols [57] according to the Guide for the Care and Use of Laboratory Animals [58]. The maintenance of zebrafish and procedures using zebrafish were approved by the Committee of Animal Care and Use of Raydel Research Institute (approval code RRI-20-003, Daegu, Republic of Korea). The fish were maintained in a system cage at 28 °C during treatment under a 10:14 h light cycle with the consumption of normal tetrabit (TetrabitGmbh D49304, 47.5% crude protein, 6.5% crude fat, 2.0% crude fiber, 10.5% crude ash, containing vitamin A [29,770 IU/kg], vitamin D3 [1860 IU/kg], vitamin E [200 mg/kg], and vitamin C [137 mg/kg]; Melle, Germany).

### 4.12. Microinjection of Zebrafish Embryos

Embryos at one-day post-fertilization (dpf) were microinjected individually using a pneumatic picopump (PV830; World Precision Instruments, Sarasota, FL, USA) equipped with a magnetic manipulator (MM33; Kantec, Bensenville, IL, USA) with a pulled microcapillary pipette-using device (PC-10; Narishigen, Tokyo, Japan). Injection of each rHDL alone (16 μg of apoA-I) or co-injection with CML (500 ng) was performed at the same position in the yolk to minimize bias, as described previously [43,59]. After the injection, the live embryos were observed under a stereomicroscope (Motic SMZ 168; Hong Kong) and photographed (20× magnification) using a Motic cam2300 CCD camera. At 24 h post-injection, each live embryo was compared after removing the chorion to compare the developmental stage at higher magnification (50×).

### 4.13. Statistical Analysis

The data are expressed as the mean ± SD from at least three independent experiments with duplicate samples. Each rHDL treatment in the in vitro studies was compared with a paired *t*-test. For the zebrafish study, multiple groups were compared using a one-way analysis of variance (ANOVA) between the groups using the Scheffe test. Statistical analysis was performed using the SPSS software program (version 28.0; SPSS, Inc., Chicago, IL, USA). A *p*-value < 0.05 was considered significant.

## Figures and Tables

**Figure 1 ijms-24-03186-f001:**
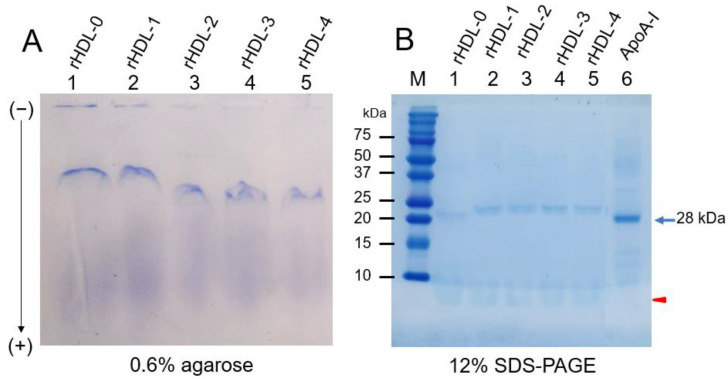
Electrophoretic profiles of rHDL containing policosanol under denatured state (**A**) and denatured state (**B**). (**A**) Native electrophoresis of each rHDL under the non-denatured state on 0.6% agarose (16 μg of protein/lane) to compare electromobility depends on the three-dimensional structure of apoA-I/HDL and its oxidation extent. The apoA-I in rHDL was visualized by Coomassie brilliant blue staining (final 1.25%). (**B**) Electrophoretic patterns of each rHDL under the denatured state on 12% SDS-PAGE (5 μg of protein/lane). The red arrowhead indicates phospholipid debris. M, molecular weight standard (Precision plus protein standards, Bio-Rad Cat # 161-0374). The gel was stained by Coomassie brilliant blue (final 0.125%) staining to visualize apoA-I and phospholipid.

**Figure 2 ijms-24-03186-f002:**
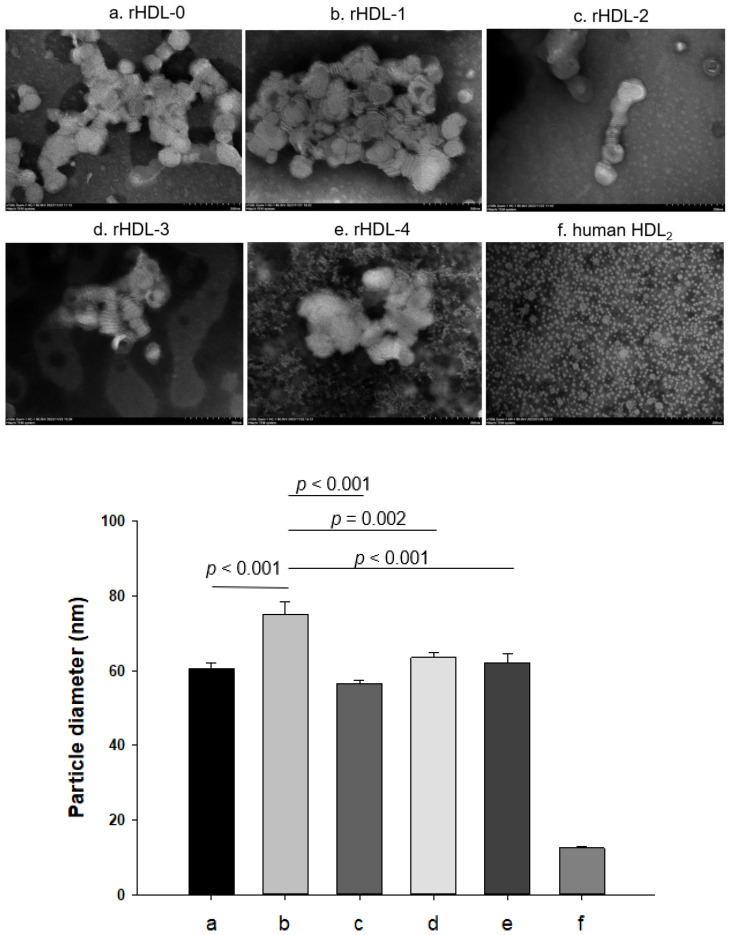
TEM image of each rHDL containing policosanol (photo **a**–**e**) and human HDL_2_ (photo **f**) with 40,000× magnification. Among rHDLs, rHDL-1 (photo **b**) showed the biggest size (as shown in graph), the most distinct discoidal particle shape with a rouleaux pattern, and the highest particle number. Human HDL_2_ from young and healthy males showed a spherical shape with a higher particle number.

**Figure 3 ijms-24-03186-f003:**
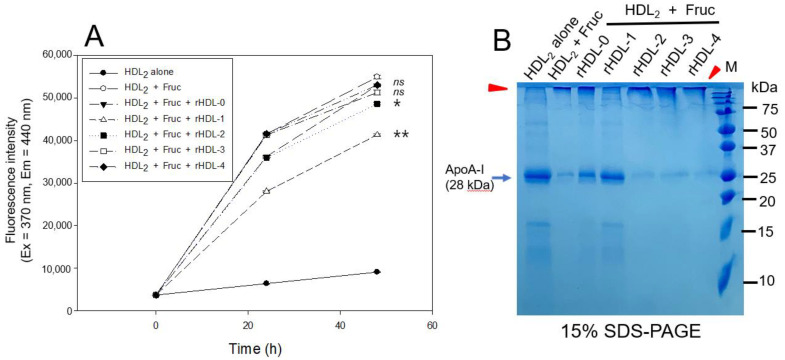
Anti-glycation activity of rHDL-containing policosanol in fructose-treated HDL_2_. (**A**) Fluorescence spectroscopic analysis (Ex = 370 nm, Em = 440 nm) of HDL_2_ (2 mg/mL of protein), which was co-treated with fructose (final 250 mM) and each rHDL (2 mg/mL of apoA-I) containing policosanol (final 3 μg/mL) during 48 h incubation. Data were expressed as mean ± SD from three independent experiments with duplicate samples. **, *p* < 0.01 versus HDL_2_ + Fruc; *, *p* < 0.05 versus HDL_2_ + Fruc; ns, not significant versus HDL_2_ + Fruc. Each rHDL treatment was compared with HDL_2_ + Fruc by paired *t*-test. (**B**) Electrophoretic patterns of the HDL_2_ (5 μg/lane) after incubation with fructose and each rHDL (15% SDS-PAGE).

**Figure 4 ijms-24-03186-f004:**
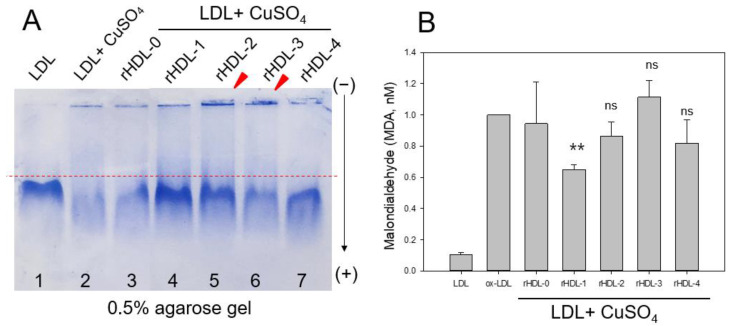
Comparison of the antioxidant abilities of rHDL-containing policosanol during cupric-ion-mediated LDL oxidation. (**A**) Comparison of relative electromobility of a mixture of LDL (8 μg of protein) and each rHDL (0.5 μg of protein) under a non-natured state on 0.5% agarose gel (120 mm length × 60 mm width × 5 mm thickness). The electrophoresis was carried out with 50 V for 1 h in Tris-acetate-ethylene-diamine-tetraacetic acid (EDTA) buffer (pH 8.0). The apo-B in LDL was visualized by Coomassie brilliant blue staining (final 1.25%). (**B**) Determination of oxidation extent by TBARS assay with the malondialdehyde (MDA) standard. Each rHDL treatment was compared with LDL + CuSO_4_ (oxLDL) by paired *t*-test. Data were expressed as mean ± SD from three independent experiments with duplicate samples. **, *p* < 0.01 versus oxLDL alone; ns, not significant versus oxLDL alone.

**Figure 5 ijms-24-03186-f005:**
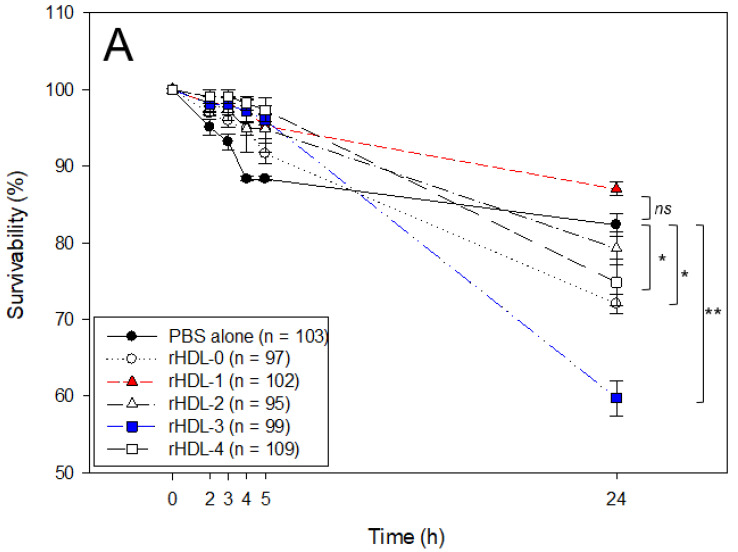
Embryo survivability and developmental morphology after the injection of rHDL containing around 16 ng of apoA-I and 250 pg of each policosanol. (**A**) Survivability of zebrafish embryo during 24 h post-injection of each rHDL. *, *p* < 0.05 versus PBS alone; **, *p* < 0.01 versus PBS alone; ns, not significant versus PBS alone. Embryo numbers were adjusted from three independent ex-periments. Statistical differences of multiple groups were compared using a one-way analysis of variance (ANOVA). (**B**) Developmental morphology of the embryos at 5 h, 24 h, 48 h, and 72 h post-injection. Red arrowheads indicate defective development and embryo death in the rHDL-3 group (photo e) at 24 h. The blue arrowhead indicates the slowest developmental speed in eye pigmentation and tail elongation in the rHDL-3 group (photo e) at 24 h-post injection. a. PBS-alone injected; b. rHDL-0 injected; c. rHDL-1 injected; d. rHDL-2 injected; e. rHDL-3 injected; f. rHDL-4 injected.

**Figure 6 ijms-24-03186-f006:**
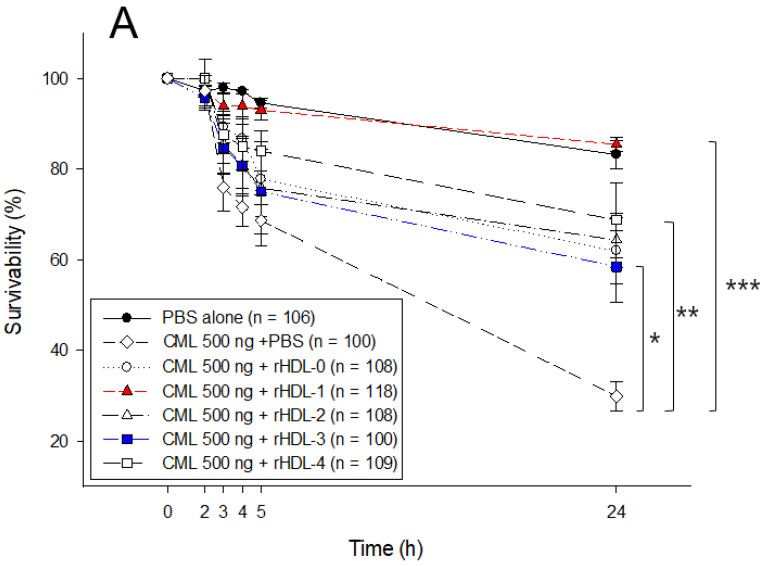
Comparison of survivability and embryo development among rHDLs containing each policosanol in the presence of carboxymethyllysine (CML). (**A**) Survivability of embryos during 24 h post-injection of each rHDL. * *p* < 0.05 vs. rHDL-3; ** *p* < 0.01 vs. rHDL-0 or rHDL-2, or rHDL-4; *** *p* < 0.001 vs. rHDL-1. Embryo numbers were adjusted from three independent experiments. Statistical differences of multiple groups were compared using a one-way analysis of variance (ANOVA). (**B**) Developmental morphology of the embryo at 5 h, 24 h, 48 h, 72 h, and 96 h post-injection. The red arrowheads indicate defected development and death of embryos in the CML alone group (photo b), CML+rHDL-2 (photo e), and CML+rHDL-3 (photo f). The blue arrowhead indicates the slowest developmental speed in eye pigmentation and tail elongation in the CML alone group (photo b) and CML + rHDL-3 (photo f) at 24 h post injection. a. PBS-alone injected; b. CML + PBS injected c. CML + rHDL-0 injected; d. CML + rHDL-1 injected; e. CML + rHDL-2 injected; f. CML + rHDL-3 injected; g. CML + rHDL-4 injected.

**Figure 7 ijms-24-03186-f007:**
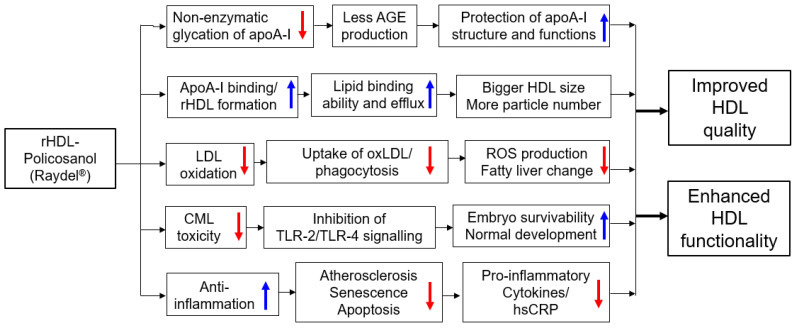
Proposed mechanism of rHDL containing policosanol (Raydel^®^) for enhancement of HDL quality and functionality via inhibition of glycation of apoA-I and oxidation of LDL. AGE, advanced glycated end product; CML, carboxymethyllysine; hsCRP, high-sensitive C-reactive protein; HDL, high-density lipoproteins; LDL, low-density lipoproteins; rHDL, reconstituted HDL; ROS, reactive oxygen species; TLR, toll-like receptor.

**Table 1 ijms-24-03186-t001:** Total wax alcohol contents and ingredient compositions from different products of policosanols.

Product Name/Description	Sugar Cane Wax Alcohol, Policosanol 1		Sugar Cane, Policosanol 2	Sugar Cane, Policosanol 3	Rice Bran, Policosanol 4
Country	Cuba		China	China	China
Manufacturer	CNIC		Xi’an Natural	Xi’an Realin	Shaanxi
Source	Sugar Cane Wax		Sugar Cane	Sugar Cane	Rice Bran
Ingredients of Long-Chain Aliphatic Alcohols	Desirable Range ^1^ (mg/g)	Determined Amount (mg/g)	Determined Amount (mg/g)	Determined Amount (mg/g)	Determined Amount (mg/g)
Total amount on the label	>900	982	600	700	980
1-tetracosanol (C24)	0.1–20	0.3	7	56	0.1
1-hexacosanol (C26)	30–100	38	89	95	5
1-heptacosanol (C27)	1–30	9	9	2	5
1-octacosanol (C28)	600–700	692	356	69	492
1-nonacosanol (C29)	1–20	6	12	8	2
1-triacontanol (C30)	100–150	139	132	236	12
1-dotriacontanol (C32)	50–100	78	3	nd	nd
1-tetratriacontanol (C34)	1–50	20	0.1	49	nd
Determined final total amount (mg)	more than 900	982	610	592	518

^1^ adopted from [28]; nd, not detected; CNIC, National Center for Scientific Research (CNIC), Habana, Cuba.

**Table 2 ijms-24-03186-t002:** Characterization of rHDL containing policosanol from different sources.

Name	Description	MW of PCO (Averaged)	Molar Ratio POPC:FC:apoA-I:PCO	WMF (nm)	Diameter (nm)
rHDL-0	rHDL alone	-	95:5:1:0	332.0	60.6
rHDL-1	Policosanol 1-rHDL	418.0	95:5:1:1	330.1	75.1
rHDL-2	Policosanol 2-rHDL	412.2	95:5:1:1	330.8	56.3
rHDL-3	Policosanol 3-rHDL	426.1	95:5:1:1	330.9	63.6
rHDL-4	Policosanol 4-rHDL	410.8	95:5:1:1	330.7	62.2

PCO, policosanol; MW, molecular weight (averaged); POPC, palmitoyloleoyl phosphatidylcholine; FC, free cholesterol; WMF, wavelength maximum fluorescence.

## Data Availability

The data used to support the findings of this study are available from the corresponding author upon reasonable request.

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
