# Peer review of "Cuban Sugar Cane Wax Alcohol Exhibited Enhanced Antioxidant, Anti-Glycation and Anti-Inflammatory Activity in Reconstituted High-Density Lipoprotein (rHDL) with Improved Structural and Functional Correlations: Comparison of Various Policosanols"

_ijms, 2023, doi:10.3390/ijms24043186_

Round 1

Reviewer 1 Report

Comments are in the document

Author Response

Thank you very much for the valuable reviewing to improve our paper.

All the comments are reflected as attached doc.

Reviewer 2 Report

Policosanols have been gaining attention of researchers and patients as lipid- and blood pressure lowering supplements. However, despite the increasing number of human studies, policosanols have not been standardized which makes the study results difficult to compare and could explain discrepancies in the published data. Thus, the presented study showing the significant differences in composition and functionality of the tested policosanols opens a new venue in this research area.

I have no major comments.

Author Response

Thank you very much heartily for reviewing and comprehension to improve our paper.

Reviewer 3 Report

This study investigated the morphological changes of rHDL, including particle size, antioxidant ability against LDL oxidation, anti-glycation activity to protect apoA-I from degradation, and anti-inflammatory activity to prevent embryonic death in the presence of CML, which is very interesting and important in the field of lipid metabolism and atherosclerosis. We believe that this research is very interesting and important in the field of lipid metabolism and atherosclerosis.

In this study, rHDL was created by incorporating polycosanol into lipoproteins, and then its physiological activity was examined. As the authors describe, it is understandable that this approach is necessary because polycosanol is difficult to dissolve in water. Is it possible to create a technique to study polycosanol by making a complex with high concentrations of BSA? Or is it not possible to analyze a complex of BSA and policosanol with normal rHDL?

The authors should represent that in the electrophoresis of Fig. 1A and Fig. 4A, the presence and location of ApoA-I should be revealed by western blotting after transfer to PVDF membrane, not just Gel-only presentation.

Figure 4A is presumably the result of agarose electrophoresis, but the details of the electrophoresis are not described. The details, including gel concentration, should be included in figure legends.

It may be better to consider in more detail what components may contribute to the significance of Cuban policosanol compared to other policosanols.

Author Response

Thank you very much for the valuable reviewing to improve our paper.

All the comments are reflected on the revision as attached doc.

Reviewer 4 Report

In this study, Cho and colleagues tested the effects of policosanols (PCO) from different origins (Cuban, Chinese), in association with reconstituted high-density lipoproteins (rHDLs), given that these compounds are poorly soluble in aqueous solutions, in different in vitro an in vivo settings (zebrafish). They report that HDLs reconstituted with cuban PCO display the strongest effects on all parameters studied, including antioxidant, anti-inflammatory, anti-glycation tests, as well as protective effects on zebrafish embryo development in normal or pro-inflammatory stress conditions. The fact that PCO-enriched rHDLs display improved beneficial effects relative to rHDLs alone is already an important result. 

Major concerns:

The comparison with differents sources of PCO for HDL enrichment needs to be reinforced by the quantification of the different components contained in rHDLs after reconstitution. Indeed the best effects obtained for the Cuban PCO may be due to a greater final PCO concentration in rHDLs and not to differences in the quality of the compound.

The size of rHDL particles obtained seems rather important relative to the known size of plasma HDL particles. Could you discuss or provide TEM microphotographs including plasma HDLs as control?

Discussion: please discuss how PCO are metabolized after ingestion. What is known about their bioavailability in humans? 

Minor:

How do you determine that the optimal and desirable content of 1-octacosanol should be around 692mg/g?

The legends of the figures are to be completed. Information is missing about the number of measurements and the statistical tests when they are carried out.

Table 1. Could you provide the annotated mass spectrum with the different lipid identifications?

Figure 1A: A western blot would be helpful to verify the presence of Apo A-I aggregates or fragments.

Figure 3: Is this a synergistic anti-glycemic effect of Apo A-I and policosanol or could the effect be due to policosanol alone?  What statistical test was performed for figure 3A? How do you explain the different Apo A-I levels in rHDL + HDL2+Fruc? An anti-ApoA-I western-blot would be helpful. 

Figure 4: Could you perform a statistical analysis on TBARS? 

Did you check the purity of LDL before performing the oxidation test? 

Figure 5: Legend to be completed. How do you explain the different number of larvae for the different experiments.

Materials and methods: The incubation time for ApoA-I glycation should be clarified (48 or 72h?). Discrepancies appear in the results vs Mat and Med.

Please use the right template of IJMS (not Acoustics as it is in the present manuscript).

Author Response

Thank you very much for the valuable reviewing to improve our paper.

All the comments are reflected on the revision as attached doc 
